# When Does Self-supervision Improve Few-shot Learning? - A Reproducibility Report

## Reproducibility Summary

**Scope of Reproducibility**

The paper investigates applying self-supervised learning (SSL) as a regularizer to meta-learning based few-shot learners. The authors claim that SSL tasks reduce the relative error of few-shot learners by 4% - 27% even when the datasets are small, and the improvements are greater when the amount of supervision is lesser or the task is more challenging. Further, they observe that incorporating unlabelled images from other domains for SSL can hurt the performance, and propose a simple algorithm to select images for SSL from other domains to provide further improvements.

**Methodology**

We reimplement the algorithms in PyTorch, starting with the author's codebase as a reference. We had to correct several bugs in the author's codebase, and reimplement the domain selection algorithm from scratch since the codebase did not contain it. We conduct experiments involving combinations of supervised and self-supervised learning on multiple datasets, on 2 different architectures and perform extensive hyperparameter sweeps to test the claim. We used 4 GTX 1080Ti GPUs throughout, and all our experiments including the sweeps took a total compute time of 980 GPU hours.

**Results**

On the ResNet-18 architecture and an image size of 224 that the paper uses throughout, our results on 6 datasets overall verify the claim that SSL regularizes few-shot learners and provide higher gains with difficult tasks. Further, our results also verify that out-of-distribution images for SSL hurt the accuracy, and the domain selection algorithm that we implement from scratch also verifies the paper's claim that the algorithm can choose images from a large pool of unlabelled images from other domains, and improve the performance.

Going beyond the original paper, we also conduct SSL experiments on 5 datasets with the Conv-4-64 architecture with an image size of 84, and find that self-supervision *does not* help boost the accuracy of few-shot learners in this setup. Further, we also show results on a practical real-world benchmark on *cross-domain few-shot learning*, and show that using self-supervision when training the base models degrades performance when evaluated on these tasks.

**What was easy**

The paper was well written and easy to follow, and provided a clear description of the experiment. The author's code implementations were relatively easy to understand and mostly reflected the experiments described in the paper.

**What was difficult**

Since the codebase was not fully complete, it took us a lot of time to identify and solve bugs, and reimplement the algorithms not present in the code. Further, multiple datasets needed a lot of preprocessing to be used. The number of hyperparameters being too many but each proving to be important, and evaluating all the claims of the paper on 5 datasets and 2 architectures was difficult to the number of experiment configurations, resulting in a very high computational cost of 980 GPU hours.

**Communication with original authors**

We maintained contact with the authors throughout the challenge to clarify several implementation details and questions regarding the domain selection algorithm. The authors were responsive and replied promptly with detailed explanations.

# 1   Introduction

Deep learning has made major advances, however this has been possible only due to the availablity of large annotated datasets for each task. Methods such as data augmentation and regularization alleviate overfitting in low-data regimes, but not completely. This motivated research in few-shot learning, in which we aim to build a classifier that should be adapted to learn new classes not seen in training, with very few samples in each class. In this work, we reproduce the paper "When Does Self-supervision Improve Few-shot Learning?" by Su et al. [13] (henceforth referred to as "the original paper" or "the paper") which investigates using self-supervised learning (SSL) in such low-data regimes to improve the performance of meta-learning based few-shot learners.

# 2   Scope of reproducibility

The paper claims that

- With *no additional training data*, adding self-supervised tasks such as jigsaw and rotation as an auxiliary task improves the performance of existing few-shot techniques on benchmarks across several different domains
- The benefits of self-supervision *increase* with the difficulty of the task, for example when training with a base dataset with less labelled data, or with images of lesser quality/resolution
- Additional unlabelled data from dissimilar domains, when used for self-supervision, negatively impacts the performance of few-shot learners
- The proposed domain selection algorithm can alleviate this issue by learning to pick images from a large and generic pool of images

We thoroughly reproduce all the experiments, and investigate whether the claims hold true, with the model and the six benchmark datasets used by the authors. Beyond the paper, we find that the results are biased towards the architecture used, and demonstrate that the gains do not hold when the input image size and architecture differ from those reported in the paper. We also report results on the more practical cross-domain few-shot learning setup, where we find that self-supervision does not help ImageNet-trained few-shot learners generalize to new domains better.

# 3   Methodology

The goal of a few-shot learner is to generalize is to learn representations of base classes that lead to good generalization on novel classes. To this end, the proposed framework combines *meta learning* approaches for few-shot learning with *self-supervised learning*. In general, learning consists of estimating functions $f$, the feature extractor and $g$, the classifier that minimize the empirical loss $\ell$ over the training data from base class $D_s = \{(x_i, y_i)\}_{i=1}^n$ consisting of images $x_i \in \mathcal{X}$ and labels $y_i \in \mathcal{Y}$, along with suitable regularization $\mathcal{R}$. This can be written as:

$$L_s = \sum_{(x_i, y_i) \in D_s} \ell(g \circ f(x_i), y_i) + \mathcal{R}(f, g)$$

In the original paper, the meta-learning based prototypical networks (ProtoNet) are used as part of the supervised loss. During meta-training, the ProtoNet computes the mean of the embeddings of all samples in a class. Then, a distance metric such as Euclidean distance or cosine distance is used to classify every query sample into one of the classes, using the distance from the class-prototypes. The loss over the query samples is backpropagated to the network, and this procedure is repeated for multiple episodes with $n$ randomly sampled classes in each episode, with $k$ examples in each class, hence referred to as the n-way k-shot setup. Hence the network meta-learns to provide useful class-prototypes from very few examples. At meta-test time, class prototypes are recomputed for classification and query examples are classified based on the distances to the class prototypes.

Apart from the supervised losses, the paper uses self-supervised losses $\ell_s s$ that are based on data $(\hat{x}, \hat{y})$ whose labels can be derived automatically without any human labelling:

$$L_{ss} = \sum_{(x_i) \in D_{ss}} \ell(h \circ f(\hat{x_i}), \hat{y_i})$$

The *jigsaw* task splits an image into 9 regions (3x3) and permutes the parts to obtain the input $\hat{x}$. The target label $\hat{y}$ is the index of the permuatation. The total number of indices are 9! which is reduced to 35 indices [cite - 41] by grouping the possible permutations to control the difficulty of the task.

78  The *rotation* task rotates the image by an angle $\theta \in 0°, 90°, 180°, 270°$ to obtain $\hat{x}$, with $\hat{y}$ being the index of the angle.

79  The paper uses a weighted combination of the two losses $L = (1-\alpha) * L_s + (\alpha) * L_{ss}$. The paper studies self-supervised
80  learning as a regularizer for representation learning, in the context of few-shot learning tasks.

81  The author also propose an algorithm to select images from a large-dataset for self-supervision when $D_s$ and $D_{ss}$ are
82  different. Here, a classifier is trained to distinguish the ResNet-101 features of images from $D_s$ and images from $D_{ss}$,
83  and the top-$k$ images according to the ratio $p(x \in D_s)/p(x \in D_p)$ are selected for self-supervision.

## 4  Experimental settings

### 4.1  Details regarding the code

86  The authors provide a public implementation of the code[1], which is built upon a popular codebase [2] from Chen et al [1].
87  We find that there are a lot of errors and bugs in the code, which took a lot of time to debug. This took up a considerable
88  part of our time. Further, the code for the domain selection algorithm was not present, and hence we had to *reimplement*
89  *it from scratch*. Our code [3] reuses multiple files from the original codebase, corrects several errors, provides easier
90  interfaces to train and test models, and also provides an implementation of the domain selection algorithm. We also
91  provide interfaces to train models with a different architecture, and to evaluate models in a cross-domain setup.

### 4.2  Model descriptions

93  The authors use a well-known architecture ResNet-18 for their experiments. The ResNet18 gives a 512-dimensional
94  feature for each input. For the jigsaw task, a single fully-connected (fc) layer with 512-units is added on top. Nine
95  patches of an image give nine 512-dimensional feature vectors, which are concatenated, and projected to 4096
96  dimensions using an fc layer, and then to a 35-dimensional output using another fc layer, corresponding to the 35
97  permutations for the jigsaw task.

98  For rotation prediction task, the 512-dimensional output of ResNet-18 is passed through three fc layers consecutively
99  with 128, 128, 4 units. The 4 predictions of the last layer correspond to the four rotation angles. Between each fc layer,
100  a ReLU activation and a dropout layer with a dropout probability of 0.5 are added. We leave this dropout probability as
101  is, as it would result in too many hyperparameters that we would not have been able to optimize for every experimental
102  setup.

103  Apart from the ResNet-18 architecture used in the paper, we use another architecture that is equally adapted in many
104  few-shot learning papers (1) (12) (14) (3), the Conv-4-64 architecture, which is a simpler architecture with 3x3 kernel
105  size and 64 filters at each layer. A similar extension is made for the jigsaw and rotation tasks. In multiple works in
106  the literature, this architecture has been used to process 84 x 84 images, while the ResNet variants have been used to
107  process 224 x 224 images. We follow the works and report results with the respective image sizes for each architecture.
108  Both the architectures are represented diagrammatically in table 16 and table 15 respectively in the appendix.

### 4.3  Datasets

110  Following the few-shot setup, each dataset is split into three disjoint sets, each having a different set of classes. A
111  model is trained on the *base* set, validated on the *validation* set, and tested on the *test* set. Following the paper, we
112  experiment with multiple datasets across diverse domains and denote the number of classes in the *base, val, test* splits
113  inside brackets: CUB-200-2011 (2)(64, 12, 20) , Stanford Cars (6) (98, 49, 49), FGVC-aircraft (9) (50, 25, 25), Stanford
114  dogs (5) (60, 30, 30), Oxford flowers (10) (51, 26, 26). These 5 datasets are henceforth referred to as "the smaller
115  datasets". Apart from these, we also experiment with a benchmark dataset for few-shot learning, the miniImageNet
116  dataset (16) (64, 16, 20). The original paper also reports results on Tiered-ImageNet, but we could only work with
117  miniImageNet due to compute and time constraints.

118  We use the same base-validation-novel class split as the paper, which they provide in their official repository. Each class
119  contains 3 files, one for each in base,val and novel, and lists the classes to be used, along with all the image paths for
120  each class. These files follow from the repository of Chen et al (1) whose codebase they borrow.

---

[1]https://github.com/cvl-umass/fsl_ssl
[2]https://github.com/wyharveychen/CloserLookFewShot
[3]https://github.com/ashok-arjun/fsl_ssl_working/

Among the small datasets, we found that there were no versions of flowers and cars dataset that could be used directly. Hence we had to preprocess the two datasets and contribute them to Kaggle for public use [4] [5]. With the miniImageNet dataset, we found that all the directly-downloadable versions ([11]) ([8]) contained images resized to 84x84, however we needed a dataset that could be resized to either 84x84 or 224x224 adaptively. Hence, we had to download the ImageNet dataset (155 GB) and process the dataset from scratch, which caused storage issues and also took up a significant part of our time. To this end, we also open-source the miniImageNet dataset with image sizes same as that in ImageNet, to save other researchers' time in preprocessing the dataset from scratch [6]. To the best of our knowledge, we are the first to release such a version.

For the domain selection algorithm, the authors use the training sets of two large datasets - Open Images v5 ([7]) and iNaturalist ([15]), which are 500 GB and 200 GB in size respectively. These sizes far exceeded our storage capacity, and we instead could only use the validation sets of each of the datasets as unlabelled images for self-supervision.

## 4.4 Hyperparameters

We perform hyperparameter sweeps each having 10 runs, amounting to 130 runs in total. The hyperparameter sweeps were conducted using Weights and Biases. Each sweep uses **random search** to search over two hyperparameters:

- **Learning Rate**: $uniform(0.0001, 0.03)$
- **Batch normalization mode**:
    1. Use batch normalization, accumulate statistics throughout training, and use the statistics during testing
    2. Use batch normalization, but do not track the running mean and variance during training; estimate them from batches during training and test
    3. No batch normalization
- $\alpha$, the weightage of the SSL term in the loss (only where self-supervision is applied)

We use these modes for batch-norm, as the paper (Page 21, Appendix A.5) states that especially for jigsaw tasks, the authors found batch-norm mode 2 to be optimal, as in jigsaw, the inputs contain both full-sized images and small patches, which might have different statistics. To verify this and for completeness, we conducted the search over the batch normalization modes also. All models are trained with the Adam optimizer with $\beta_1 = 0.9$ and $\beta_2 = 0.999$.

We then use the configuration which gives the best *validation accuracy* computed for 100 epochs, computed over 600 randomly sampled episodes. We search hyperparameters for certain datasets only, and reuse the hyperparameters found for similar datasets due to computational constraints. The selected experiment configurations are given in the appendix due to space constraints.

Across 100% of our sweeps, we notice that $\alpha$ stays below 0.6, and does not go below 0.3 in our runs. Hence, we infer that an adequate amount of supervision is also needed for good performance, and too much self-supervision hurts accuracy. For the miniImageNet datset, we find the values close 0.3 work the best, which the paper reiterates. The paper reports that they use 0.5 for all the SSL experiments on the small datasets, which we confirm as our $\alpha$ term converges to values 0.4 and 0.6 for the small datasets. All of our reported results are with the best hyperparameters found. We report more details on the hyperparameter searches in Appendix.

## 4.5 Computational requirements

We used 4 Nvidia 1080Ti GPUs for all experiments. The run-times differ for each experiment configuration when incorporating self-supervision. We report the average epoch time for each experimental setup (1 epoch = 100 episodes) in table 6 in the appendix.

In general, among experiments involving self-supervised learning, rotation took the maximum amount of time. This is because 4 rotations of the same image are needed at every instance, which is more expensive than loading a single image. The jigsaw task took lesser time than rotation, and the combination of jigsaw and rotation took the highest amount of time per epoch. Since the paper reports results on the combination only for the first set of experiments (claim 1), we also do the same. Further, the computational time restricted us from performing more experiments combining the two.

---

[4] https://www.kaggle.com/arjun2000ashok/vggflowers/
[5] https://www.kaggle.com/hassiahk/stanford-cars-dataset-full
[6] https://www.kaggle.com/arjunashok33/miniimagenet

In total, apart from the hyper-parameter sweeps, we perform **250 experiments**, across different experimental setups and multiple datasets. All of these experiments took approximately **700 GPU hours**. Along with the hyperparameter sweeps which were lesser in duration, the experiments took approximately **980 hours** of compute time.

## 4.6 Experimental setup and code

Following the authors, we train, evaluate and report results on the 5-way 5-shot setting; we also explore 20-way 5-shot setting but we could not continue after a few runs, restricted by the large training and testing time of 20-way 5-shot models. Following the paper, we use 16 query examples to evaluate the models.

On verifying that the core claim of the paper (claim 1) for all the 5 small datasets, we choose 2 to 3 representative datasets for other experiments - CUB, dogs (representing natural images) and cars (representing the other group). We could not perform all the experiments on all the 5 datasets due to computational constraints. For the domain selection, we evaluate on all the 5 datasets to verify our implementation of the algorithm.

The batch size cannot be set in episodic few-shot learners, and are by default $n\_way * (n\_support + n\_query)$. We use 16 query images following the paper, and as a result, our batch sizes are 105 in 5-way 5-shot experiments, and 420 in 20-way 5-shot experiments. Following all previous work in few-shot learning, we sample 100 episodes (batches) per epoch, and conduct experiments on about 600 - 800 epochs. Following the paper, we use only 5 query images when training models for experiments that use lesser labelled data since the {20, 40, 60, 80}% splits of dataset do not contain 16 query images in all classes.

In every iteration, an equal number of unlabelled images are sampled at random from the respective dataset(s) for self-supervised learning. Following our paper and the baseline from previous work (1) in few-shot learning and our original paper, we use the following data augmentation: For label and rotation predictions, images are first resized to 224 pixels for the shorter edge while maintaining aspect ratio, from which a central crop of 224 is obtained. For jigsaw puzzles, a random crop of 255 is done from the original image with random scaling between [0.5, 1.0], then split into 3×3 regions, from which a random crop of size 64×64 is picked.

We implement the domain selection algorithm following the paper: For each dataset among the small datasets, we select negative images uniformly at random with 10 times the size of the positive images. The loss for the positive class is scaled by the inverse of its frequency to account for the significantly larger number of negative examples. We then train a binary logistic regression classifier using LBFGS for 10000 iterations and use the logits to compute the ratio $p(x \in D_s)/p(x \in D_p)$. We then choose $k$ as 80% of the total dataset size, and sample $k$ negative images to use as unlabelled samples.

For evaluation at meta-test time, we use 600 randomly sampled episodes, and report the mean accuracy and 95% confidence intervals. Due to the large number of experiments and the datasets across which the claims had to be verified, we could only perform one set of experiments in all sections, with the seed is set to 42.

# 5 Results

## 5.1 Results reproducing original paper

Here, we consider the same architecture that the paper uses - ResNet-18, with an input image size of 224.

### 5.1.1 Self-supervision improves few-shot learning

Here, we successfully verify claim 1 of the paper that with no additional unlabelled data, SSL improves few-shot learning when applied as an auxiliary task. We conduct experiments across all the 5 small datasets as well as the large-scale miniImageNet dataset. We also reproduce results on the baseline from (1), and MAML and MAML+Jigsaw. We could not reproduce results on MAML due to computational constraints. We present results in Figure 1, Table 1 and Table 2. All results are on 5-way 5-shot classification. We find that the jigsaw task leads to the best results on 3 out of 6 datasets.

### 5.1.2 The benefits of self-supervision increase with the difficulty of the task

We successfully verify claim 2 of the authors that the relative gains of using SSL are more when the difficulty of the task is higher. The authors experiment with two types of difficult tasks: one with low-resolution/greyscale images as input, and another with less labelled data from the base training set. We experiment with 3 selected datasets and were successful in reproducing the results. We report the results on figures 2 and 4. The exact numbers are given on tables 12

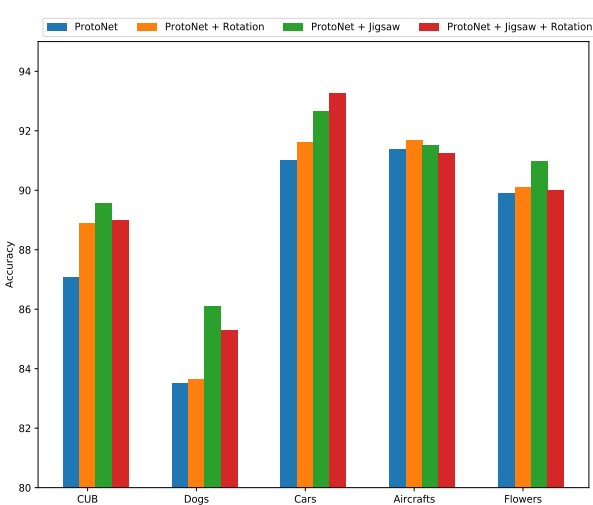

Figure 1: Results on applying SSL tasks to Prototypical networks, across 6 datasets

| Method | Accuracy |
|---|---|
| ProtoNet | 74.07 ± 0.71 |
| ProtoNet + Jigsaw | **77.29 ± 0.73** |
| ProtoNet + Rotation | 74.93 ± 0.9 |
| ProtoNet + Jigsaw + Rotation | 76.23 ± 0.9 |

Table 1: miniImageNet Results with ResNet-18

| Method | CUB | Cars | Aircrafts | Dogs | Flowers |
|---|---|---|---|---|---|
| Softmax | 81.92 ± 0.54 | 88.16 ± 0.47 | 89.57 ± 0.38 | 78.18 ± 0.56 | 90.44 ± 0.47 |
| Softmax + Jigsaw | 83.96 ± 0.52 | 91.2 ± 0.49 | 89.93 ± 0.39 | 78.3 ± 0.57 | 90.85 ± 0.49 |
| ProtoNet | 87.09 ± 0.48 | 91.0 ± 0.41 | **91.90 ± 0.35** | 83.52 ± 0.54 | 89.92 ± 0.51 |
| ProtoNet + Jigsaw | **89.57 ± 0.43** | 92.67 ± 0.39 | 91.72 ± 0.39 | **86.1 ± 0.51** | **90.98 ± 0.47** |
| ProtoNet + Rotation | 88.9 ± 0.55 | 91.61 ± 0.40 | 91.69 ± 0.40 | 83.94 ± 0.58 | 90.12 ± 0.5 |
| ProtoNet + Jigsaw + Rotation | 88.98 ± 0.45 | **93.27 ± 0.38** | 91.26 ± 0.4 | 85.29 ± 0.54 | 90.01 ± 0.51 |

Table 2: ResNet-18's performance on the 5 small datasets.

and 10 respectively in the appendix. We find that the claims of the paper hold true, and that self-supervision has higher gains in harder tasks.

### 5.1.3 Unlabelled data for SSL from dissimilar domains negatively impacts the few-shot learner

Verifying claim 3 of the paper, we replace a portion of the labelled data, starting from 20% of the data to 80% of the data, with data from other domains. Here, we combine the data from all other datasets together, and sample images at random. We present results on 3 chosen datasets, again, to save computation and time for other results. Results are given in figure 3 and table 11 (appendix). The claim that using data from dissimilar domains for self-supervision is detrimental to few-shot classification holds true.

### 5.1.4 The proposed domain selection algorithm can alleviate this issue by learning to pick images from a large and generic pool of images

To verify claim 4, we implement the domain selection algorithm from scratch, and verify it across all 5 small datasets as given in the paper, to make sure that we have got the implementation right. Results are presented in 5 and table 13 in the appendix. Results are shown on using only 20% of the labelled data for learning, only selecting images from other domains at random, and on using the proposed domain selection algorithm. We successfully verify and demonstrate that the algorithm proposed by the authors for selecting images from multiple dissimilar domains.

## 5.2 Results beyond original paper

### 5.2.1 Results on a different architecture - Conv4

Here, we aim to investigate whether the claims of the paper hold when a small architecture that needs a smaller image size (84x84) is used. In particular, we investigate claim 1 of the paper extensively. Note that the authors do not report results with this architecture. Results are given in figure 6 and table 7 and table 9 (appendix). We find that the results **do**

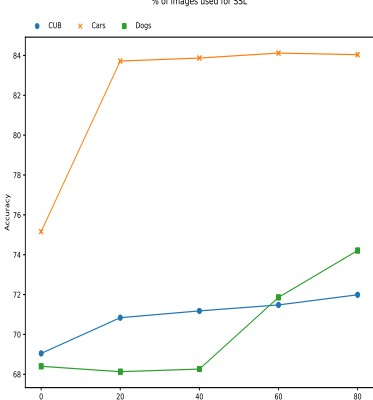

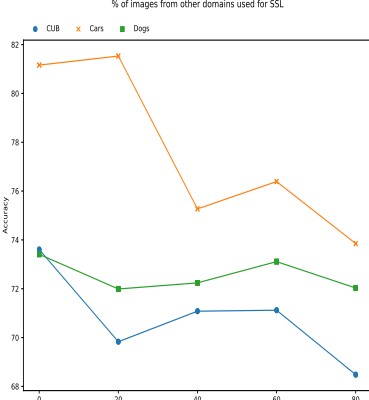

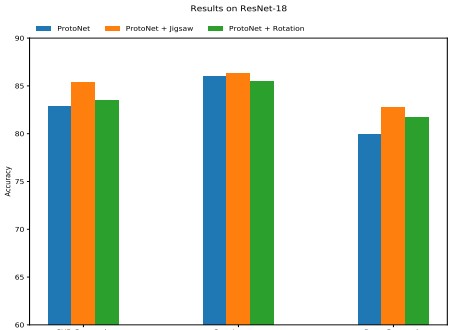

Figure 2: Results of applying SSL when the amount of labelled data for supervision is lesser. The gains obtained by SSL grow with the amount of labelled data

Figure 3: Performance on tasks where a portion of the labelled data is replaced with data from other domains

Figure 4: Results of applying self-supervised learning on artificially constructed harder tasks.

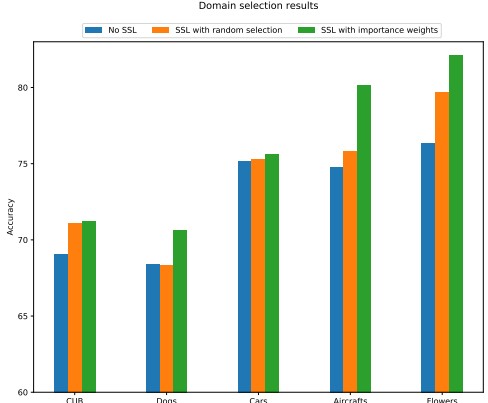

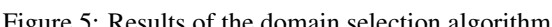

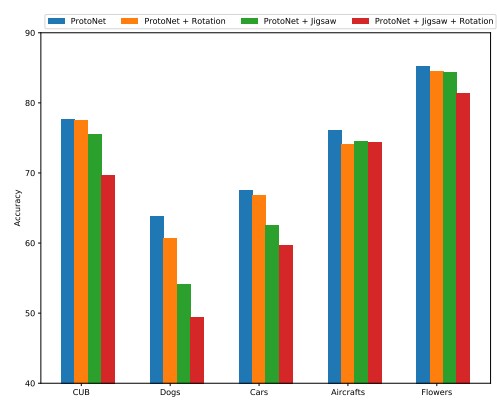

Figure 5: Results of the domain selection algorithm

Figure 6: Results of using SSL with the Conv4 architecture

**not hold true** when a smaller architecture and image size is used, and that claim depends heavily on the architecture and image size. We present results across all the 5 small datasets for completeness, across both SSL tasks. To confirm our claims, we also rerun results with another seed, but get similar results (Table 8 in appendix). Apart from the reported results with the optimal $\alpha$ found by hyperparameter search, we study the effect of $\alpha$ on the results, with the CUB and cars datasets in tables 3 and 4. Here we find that the value of $\alpha$ plays an important role in the performance, and that high values cause too much supervision when the model is small. Even across training and testing with multiple $\alpha$ values, we find that the self-supervision provides only a marginal boost in 1 out of 4 cases, invalidating claim 1 of the paper that self-supervision provides a stable boost to few-shot learners.

| Rotation | CUB | Cars |
|---|---|---|
| $\alpha = 0$ (no SSL) | **77.72 ± 0.71** | **67.6 ± 0.84** |
| $\alpha = 0.1$ | 77.6 ± 0.73 | 66.83 ± 0.75 |
| $\alpha = 0.3$ | 77.22 ± 0.9 | 65.53 ± 0.73 |
| $\alpha = 0.5$ | 75.04 ± 0.81 | 60.74 ± 0.73 |

Table 3: Conv-4's performance on Rotation

| Jigsaw | CUB | Cars |
|---|---|---|
| $\alpha = 0$ (no SSL) | **77.72 ± 0.71** | **67.6 ± 0.84** |
| $\alpha = 0.1$ | 75.57 ± 0.73 | 62.548 ± 0.75 |
| $\alpha = 0.3$ | 64.91 ± 0.9 | 51.83 ± 0.73 |
| $\alpha = 0.5$ | 75.04 ± 0.81 | 60.74 ± 0.73 |

Table 4: Conv-4's performance on Jigsaw

### 5.2.2 Results on cross-domain few-shot learning

In another effort to extend the paper's results, we test the results of our trained models on the BSCD-FSL benchmark for cross-domain few-shot learning, introduced by (4) with their code [7]. The benchmark requires ImageNet-based trained few-shot models to evaluated on four cross-domain datasets: CropDiseases, EuroSAT, ISIC2018, and ChestX datasets, which covers plant disease images, satellite images, dermoscopic images of skin lesions, and X-ray images, respectively. The selected datasets reflect real-world use cases for few-shot learning since collecting enough examples from above domains is often difficult, expensive, or in some cases not possible. We use this benchmark to find out if models trained with self-supervision provide gains over normal supervised models when tested on *real-world datasets*. We test our mini-ImageNet trained models on this benchmark, to find out if self-supervision improves results on cross-domain datasets. Results on the ResNet-18 models are reported in table 5. Results on the Conv-4 models are deferred to the appendix table 14. We find that self-supervision results in learning heavily domain-specific representations, and that the results of the fully-supervised learner are much better than those with auxiliary tasks as self-supervision.

| Method | ChestX | Crop Disease | EuroSAT | ISIC |
|---|---|---|---|---|
| ProtoNet | **24.32 ± 0.41** | **83.36 ± 0.63** | **76.09 ± 0.74** | 41.60 ± 0.58 |
| ProtoNet + Jigsaw | 23.97 ± 0.39 | 77.86 ± 0.69 | 72.72 ± 0.68 | 41.22 ± 0.56 |
| ProtoNet + Rotation | 23.84 ± 0.39 | 79.11 ± 0.68 | 72.47 ± 0.69 | **43.79 ± 0.61** |
| ProtoNet + Jigsaw + Rotation | 23.73 ± 0.38 | 77.39 ± 0.68 | 71.91 ± 0.7 | 40.05 ± 0.55 |

Table 5: CDFSL Benchmark for ResNet-18

## 6 Discussion

We find that the **central claims of the author as given in Section 2 hold true, when the same architecture is used**. Considering the ResNet-18 model used in the paper with an input image size of 224, we find that self-supervision - in particular the jigsaw task, provides a boost in the case of small datasets. Experimentally, we verify claim 1 of the paper on all small datasets and miniImageNet. However, going beyond the paper's architecture, we find that the results depend heavily on the image size and architecture and do not give the same gains with Conv-4-64, another architecture common in the few-shot learning literature, with an input image size of 84. Further ablation reveals that the jigsaw task in particular has a strong influence in this setup, and the rotation task requires tuning the $\alpha$ parameter to even reach the accuracy of the fully-supervised model. Future work may investigate ways to boost the performance of few-shot classifiers when the input sizes are small, and may also find out better architectures to use when the input size is small. Future work may also experiment with other available architectures, and find out if self-supervision increases performances across all configurations.

Regarding claims 2 and 3 such as on harder tasks and scenarios with lesser labelled data in the base dataset, our experiments on selected datasets verify that the claims hold true, with the ResNet-18 backbone. Further, we verify claim 4 of the paper by implementing the domain selection algorithm from scratch and our experiments on all the 5 datasets show that relative gains are achieved. Future work may also investigate if the same claims hold true when different architectures were used.

Finally, we evaluate the miniImageNet-trained models on a more practical setting of cross-domain few-shot learning and find that SSL during the training time does not help few-shot learners generalize across domains better. Future work may investigate why applying SSL results in domain-specific features, and propose methods to apply SSL in a more domain-agnostic manner. We recommend future works in few-shot learning to train and evaluate in multiple architectures with different image sizes and verify their work more thoroughly.

### 6.1 Communication with original authors

We maintained communication with the authors throughout our implementation and training phase, spanning two months. We were able to clarify many implementation details in the original codebase, and the authors also re-ran an experiment on their side to test if the numbers match. Further, we recieved a lot of help regarding implementation of the domain selection algorithm, and could also confirm the implementation with them. We acknowledge and thank the authors for their help with the reproducibility of their paper.

---

[7]https://github.com/IBM/cdfsl-benchmark

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

# 7 Appendix

## 7.1 Seconds per epoch

Continuing section 4.5, we report the exact values per epoch across experiment configurations. We do so, since different architectures and datasets may require training for different number of epochs, however the epoch time remains the same across experiments.

| Experiment | Setup (way,shot) | Seconds per epoch (Conv4 / ProtoNet |
|:---:|:---:|:---:|
| ProtoNet | (5,5) | 20/25 |
| ProtoNet | (20,5) | 45/50 |
| ProtoNet+Jigsaw | (5,5) | 25/35 |
| ProtoNet+Jigsaw | (20,5) | 60/66 |
| ProtoNet+Rotation | (5,5) | 18/60 |
| ProtoNet+Rotation | (20,5) | 65/81 |
| ProtoNet+Jigsaw+Rotation | (5,5) | 42/70 |
| ProtoNet+Jigsaw+Rotation | (20,5) | 83/95 |

Table 6: Average seconds per epoch across experimental setups and ways

## 7.2 Hyperparameter sweeps

The selected experiment configs are as follows:

Each of the below experimental configurations are done for ProtoNet, ProtoNet+Jigsaw, ProtoNet+Rotation and ProtoNet+Jigsaw+Rotation (4 configurations) in the 5-way 5-shot setup. The sweeps optimize the **learning rate** and **the mode of batch normalization**, and $\alpha$.

The last two parameters are optimized only when self-supervision is applied. This is because $\alpha = 0$ for fully supervised learners and we find that using batch norm modes 2,3 is highly detrimental to fully supervised learners.

- miniImageNet Conv4: **4** sweeps
- miniImageNet ResNet-18: **4** sweeps
- CUB Conv4: **4** sweeps, reused for flowers and dogs datasets
- Cars Conv4: **4** sweeps, reused for aircrafs dataset
- CUB ResNet-18: **4** sweeps, reused for flowers and dogs datasets
- Cars ResNet-18: **4** sweeps, reused for aircrafs dataset

Hence we do a total of **24** sweeps.

The sweeps and the exact hyperparameters obtained can be visualized at https://wandb.ai/meta-learners/FSL-SSL/sweeps. All the runs in the paper can be seen at https://wandb.ai/meta-learners.

### 7.3 Tables

#### 7.3.1 Results on the applying self-supervision to few-shot learners

| Method | CUB | Cars | Aircrafts | Dogs | Flowers |
|---|---|---|---|---|---|
| ProtoNet | **77.72 ± 0.48** | **67.99 ± 0.41** | **76.16 ± 0.69** | **63.88 ± 0.54** | **85.29 ± 0.51** |
| ProtoNet + Jigsaw | 75.57 ± 0.7 | 62.54 ± 0.39 | 74.53 ± 0.68 | 54.27 ± 0.51 | 84.4 ± 0.47 |
| ProtoNet + Rotation | 77.5 ± 0.55 | 66.8 ± 0.40 | 74.16 ± 0.40 | 60.74 ± 0.58 | 84.55 ± 0.5 |
| ProtoNet + Jigsaw + Rotation | 69.66 ± 0.45 | 59.76 ± 0.77 | 74.79 ± 0.4 | 49.48 ± 0.54 | 81.43 ± 0.51 |

Table 7: Conv-4's performance on few-shot learning tasks ($\alpha = 0.5$)

| Method | CUB | Cars |
|---|---|---|
| ProtoNet | 76.43 ± 0.3 | 67.45 ± 0.85 |
| ProtoNet + Jigsaw | 65.09 ± 0.42 | 60.39 ± 0.76 |
| ProtoNet + Rotation | 75.05 ± 0.35 | 66.61 ± 0.6 |

Table 8: Conv4 results on CUB and cars with a different seed

| Method | Conv-4 |
|---|---|
| ProtoNet | **66.78 ± 0.84** |
| ProtoNet + Jigsaw | 64.94 ± 0.75 |
| ProtoNet + Rotation | 66.41 ± 0.73 |
| ProtoNet + Jigsaw + Rotation | 65.21 ± 0.73 |

Table 9: miniImageNet Results on Conv4

#### 7.3.2 Results on harder tasks

| Method | 20% CUB | 20% Cars | 20% Dogs |
|---|---|---|---|
| No SSL | **73.61 ± 0.71** | 75.16 ± 0.84 | 68.4 ± 0.64 |
| 20% SSL | 70.84 ± 0.73 | 83.72 ± 0.75 | 68.13 ± 0.9 |
| 40% SSL | 71.48 ± 0.9 | 83.87 ± 0.73 | 68.26 ± 0.87 |
| 60% SSL | 70.71 ± 0.81 | **84.12 ± 0.73** | **74.21 ± 0.89** |
| 80% SSL | 71.99 ± 0.65 | 84.04 ± 0.78 | 71.86 ± 0.81 |

Table 10: Performance on tasks with lesser labelled data

| Method | 20% CUB | 20% Cars | 20% Dogs |
|---|---|---|---|
| No SSL | **73.61 ± 0.82** | 75.16 ± 0.84 | 68.4 ± 0.64 |
| 20% SSL | 69.83 ± 0.79 | **81.53 ± 0.79** | 71.99 ± 0.88 |
| 40% SSL | 71.08 ± 0.83 | 75.27 ± 0.89 | 72.24 ± 0.85 |
| 60% SSL | 71.12 ± 0.91 | 76.39 ± 0.89 | **73.11 ± 0.83** |
| 80% SSL | 68.48 ± 0.87 | 73.85 ± 0.89 | 72.03 ± 0.91 |

Table 11: Performance when a portion of data replaced with data from other domains

| Method | CUB Greyscale | Cars Low-resolution | Dogs Greyscale |
|---|---|---|---|
| ProtoNet | 82.88 ± 0.56 | 86.00 ± 0.51 | 79.97 ± 0.54 |
| ProtoNet + Jigsaw | **85.44 ± 0.52** | **86.34 ± 0.56** | **82.82 ± 0.50** |
| ProtoNet + Rotation | 83.51 ± 0.55 | 85.53 ± 0.53 | 81.74 ± 0.59 |

Table 12: Performance on artificially constructed harder tasks

### 7.3.3 Results on domain selection

| Method | CUB | Cars | Aircrafts | Dogs | Flowers |
|---|---|---|---|---|---|
| No SSL | 69.05 ± 0.48 | 75.15 ± 0.41 | 74.8 ± 0.35 | 68.4 ± 0.54 | 76.34 ± 0.51 |
| SSL Pool (Random) | 71.11 ± 0.43 | 75.27 ± 0.39 | 75.81 ± 0.39 | 68.38 ± 0.51 | 79.71 ± 0.47 |
| SSL Pool (Weight) | **71.25 ± 0.55** | **75.65 ± 0.40** | **80.13 ± 0.40** | **70.66 ± 0.58** | **82.16 ± 0.5** |

Table 13: Domain selection results

### 7.3.4 Results on cross-domain few-shot learning

| Method | ChestX | Crop Disease | EuroSAT | ISIC |
|---|---|---|---|---|
| ProtoNet | **24.46 ± 0.39** | **80.45 ± 0.66** | 67.03 ± 0.7 | **41.0 ± 0.6** |
| ProtoNet + Jigsaw | 24.07 ± 0.4 | 78.51 ± 0.66 | 64.69 ± 0.7 | 39.81 ± 0.54 |
| ProtoNet + Rotation | **24.46 ± 0.39** | 79.30 ± 0.7 | 66.50 ± 0.71 | 39.54 ± 0.54 |
| ProtoNet + Jigsaw + Rotation | 24.16 ± 0.37 | 78.67 ± 0.66 | **67.60 ± 0.66** | 40.22 ± 0.54 |

Table 14: CDFSL Benchmark for Conv-4.

## 7.4 Architectures

| Layer Name | Output Size | Conv-4-64 |
|---|---|---|
| conv1 | 82 x 82 x 64 | 3 x 3, 64 |
| conv2 | 41 x 41 x 64 | 2 x 2, max pool, stride 2 |
| | | 3 x 3, 64 |
| conv3 | 18 x 18 x 64 | 2 x 2, max pool, stride 2 |
| | | 3 x 3, 64 |
| conv4 | 7 x 7 x 64 | 2 x 2, max pool, stride 2 |
| | | 3 x 3, 64 |
| average pool | 1 x 1 x 64 | 7 x 7 average pool |
| fully connected | 1024 | 64 x 1024 linear |
| fully connected | X | 1024 x X linear |
| softmax | X | |

Table 15: Conv-4 Architecture (X denotes the way)

| Layer Name | Output Size | Conv-4-64 |
|---|---|---|
| conv1 | 112 x 112 x 64 | 7 x 7, 64, stride 2 |
| conv2_x | 56 x 56 x 64 | 3 x 3 max pool, stride 2 |
| | | [3 x 3, 64; 3 x 3, 64] x 2 |
| conv3_x | 28 x 28 x 128 | [3 x 3, 128; 3 x 3, 128] x 2 |
| conv4_x | 14 x 14 x 256 | [3 x 3, 256; 3 x 3, 256] x 2 |
| conv5_x | 7 x 7 x 512 | [3 x 3, 512; 3 x 3, 512] x 2 |
| average pool | 1 x 1 x 512 | 7 x 7 average pool |
| fully connected | X | 512 x X fully connections |
| softmax | X | |

Table 16: ResNet-18 Architecture

