# OpenReview forum: "When Does Self-supervision Improve Few-shot Learning? - A Reproducibility Report"
_ML_Reproducibility_Challenge/2021/Fall — RC2021_

### Official Review · Reviewer_psc7 · 2022-03-02
**Really good report, confirms all main claims and provides additional results showing limitations of original method**

**Rating:** 9
**Confidence:** 4

**Review:**

Reproducibility Summary
----------------------------------
The summary is complete and clear enough, with details in the main report and appendix.
It accurately includes the main findings, related to the original claims in the paper, and novel findings uncovered by the reproduction.

Two small negative points:
- The "scope of reproducibility" (see next section) only summarizes the original claims
- The summary does not mention the work done to package and ensure access to the data (in addition to the code), which was a nice contribution

Scope of reproducibility
--------------------------------
That section in the summary mostly states the claims of the original article, but not the scope of _this_ report, which:
- checks the 4 main claims in the original article, on most datasets, and one of two few-shot settings (5-way 5-shot)
- validates 3 hyper-parameters
- performs the experiments on another, smaller backbone than the original paper
- measures performance on a cross-domain few-shot learning benchmark

This information is available in the other sections of the summary (Methodology and Results), as well as in the expanded version of the scope in the report on p. 2.

Code
--------
The code re-used the original author repository, with the following additions:
- correction of errors and bugs (although it is not clear if some of these bugs impacted the original results)
- implementation of the domain selection algorithm (not part of the original code release)
- addition of another architecture (backbone)

Moreover, the authors re-constituted and packaged 3 datasets (VGG Flowers, Stanford Cars, mini-Imagenet with larger images) for easier reproduction of the experiments.

Communication with original authors
-------------------------------------------------
The report mentions regular and helpful communication with the original authors.

Hyperparameter Search
---------------------------------
The report performs a reasonable hyper-parameter search over 3 hyperparameters: learning rate, trade-off between main objective and self-supervision, batch-norm mode.
The search was only done on a subset of the datasets and re-used for others, which did not impact the final performance much.

The main addition is expanding the search to the space of architectures (neural backbones), by adding a 4-layer convnet when the original article used only a resnet-18.

Ablation Study
--------------------
No ablation study was added, although extensions are performed (different backbone, cross-domain benchmark)

Discussion on results
------------------------------
The discussion section is clear and complete, it outlines the expected and unexpected challenges encountered by the team, as well as their reaction.
In particular:
- the longer time to train in the 20-way setting, leading to they reporting on 5-way only
- the resources needed for hyperparameter sweeps, leading to re-using values across datasets
- the (re-)implementation needed for parts of the code base

Recommendations for reproducibility
---------------------------------------------------
The authors provide updated code and data that facilitate further reproducibility by other teams. In this sense, they implemented these recommendation themselves.

However, the bugs and errors they found in the original code do not seem to be reported as GitHub issues, and fixes are not proposed as pull requests on the original repo. This makes it harder for the original authors to improve their code, for other teams to scrutinize the impact of these issues, and can cause further code fragmentation.

Results beyond the paper
-----------------------------------
The report expands experiments to also consider a smaller backbone, a 4-layer convnet, and shows that the benefit of self-supervised training does not extend to this architecture.

This is not unexpected, as the original article framed self-supervised training as a _regularization_ technique, and it's plausible that this simpler network does not have enough capacity to need regularization. However, this additional result it interesting in that it confirms that self-supervised training acts like a regularizer, and provides additional data for potential users of this technique.

The report also adds experiments on a cross-domain few-shot learning benchmark, showing no benefit of self-supervised training when the  target domain is too different from the pre-training one. This sheds additional light to the limits of the original approach.

Overall organization and clarity
------------------------------------------
The paper is very clear, and well organized. The split between the main report and the appendix make sense.

Minor points:
- It would have been nice to include the results reported in the original article in the same tables besides the reproduction, to avoid back-and-forth between the original article and this report.
- Some results would be tied with the best ones, whether from a proper statistical test or even just with the 95% confidence interval overlaps, yet only one result is bolded (for instance Table 2, Aircrafts, ProtoNet and ProtoNet+Jigsaw should be tied). This was an issue in the original article too, however.

Conclusion
----------------
This is a really good reproduction report, that confirms all 4 main claims of the original article, and performs additional experiments showing limitations of the method. The released code and data will also be helpful to further efforts in that direction.

---

### Official Review · Reviewer_BF1x · 2022-03-29
**Valuable corroboration of original work**

**Rating:** 8
**Confidence:** 3

**Review:**

**Summary:**
The original paper (OP) proposes using self-supervised learning (SSL) as a data-dependent regularizer in meta learning tasks. Its primary claims are as follows:
* SSL improves the performance of few-shot meta learners, even with small datasets and without using additional data
* The improvements in performance are greater if the task is more difficult
* When the data used for SSL comes from a different distribution, performance is negatively impacted
* A simple domain classification algorithm can be used to select images for SSL from an unlabeled dataset

The authors reproduce experiments verifying the key claims of the OP using multiple benchmark datasets and extend the study to new settings, examining performance with a different architecture than what is used in the OP and presenting results on cross-domain few-shot learning benchmarks.

**Strengths:**
* The paper is well written and easy to follow, providing a concise explanation of the problem setting, classes of meta-learning methods, self-supervised losses used, and the experimental setup. The suggested format for the reproducibility report (RR) was respected.
* The authors' code is well documented and contains clear instructions on running the experiments. The authors state that they had to debug multiple errors in the OP's codebase and implement the domain selection algorithm from scratch. The code can be used to evaluate performance on different architectures.
* The authors conduct an exhaustive hyperparameter search while testing the OP's claims. They find that performance is sensitive to $\alpha$, the weightage of the SSL term in the loss. While adequate self-supervision is necessary for good performance, a high $\alpha$ can hurt the accuracy.
* The paper explicitly states wherever the authors could not reproduce a certain result due to computational constraints, which I felt adds clarity to the RR.
* The authors went beyond the OP and examined using auxiliary self-supervised tasks with the Conv4 architecture, concluding that claim 1 doesn't hold true. On the cross-domain few-shot learning experiment, they find that self-supervision results in learning very domain-specific representations, often degrading performance.
* The RR authors have communicated with the OP's authors to clarify several details, adding credence to their work.
* Preprocessed versions of the *flowers*, *cars* and *miniImageNet* datasets were open-sourced by the authors, which will be of significant use to the research community.

**Weaknesses:**

I only have the following suggestions to make:
* The authors might add a few lines of their analyses for certain results. For example, the results on applying SSL tasks to prototypical networks (Fig. 1 in RR, Fig. 2 in OP) are slightly different - specifically, using both Jigsaw and Rotation seems to degrade performance. In Sec. 5.2.1, it is reported that the Conv4 architecture does not validate claim 1 - it would be valuable to have the authors' hypotheses/ideas on why this is the case.
* A few minor typos can be fixed (e.g. line 76: permuatation -> permutation, line 76:  \[cite - 41\](?),  line 224: 5 -> figure 5)

Overall, the RR is a strong validation of the OP's key claims while also identifying problem settings where the proposed methods will not be of much use.

---

### Meta-Review · Program_Chairs · 2022-04-07

**Recommendation:** Accept
**Confidence:** 1

**Metareview:**

A very solid reproduction report that confirms all of the main claims of the original paper that was surveyed. Furthermore, it performs additional experiments showing the limitations of the original method, and shares the code and details necessary to continue building upon this work.

---

### Decision · Program_Chairs · 2022-04-09

**Decision:**

Accept

**Comment:**

Following the recommendation of reviewers and meta-reviewer, the paper is accepted for ML Reproducibility Challenge 2021, and will be published in the upcoming special edition of ReScience Journal.